# The Next Generation Cognitive Security Operations Center: Network Flow Forensics Using Cybersecurity Intelligence

**Konstantinos Demertzis** [1,*] (ID)**, Panayiotis Kikiras** [2]**, Nikos Tziritas** [3]**, Salvador Llopis Sanchez** [4] **and Lazaros Iliadis** [1]

[1]  Department of Civil Engineering, School of Engineering, Democritus University of Thrace, Xanthi 67100, Greece; liliadis@civil.duth.gr
[2]  Department of Computer Science, School of Science, University of Thessaly, Lamia 35131, Greece; kikirasp@uth.gr
[3]  Research Center for Cloud Computing, Shenzhen Institutes of Advanced Technology, Chinese Academy of Sciences, Shenzhen 518000, China; nikolaos@siat.ac.cn
[4]  Communications Department, Universitat Politecnica de Valencia, Valencia 46022, Spain; salllosa@masters.upv.es
*  Correspondence: kdemertz@fmenr.duth.gr; Tel.: +30-694-824-1881

**Abstract:** A Security Operations Center (SOC) can be defined as an organized and highly skilled team that uses advanced computer forensics tools to prevent, detect and respond to cybersecurity incidents of an organization. The fundamental aspects of an effective SOC is related to the ability to examine and analyze the vast number of data flows and to correlate several other types of events from a cybersecurity perception. The supervision and categorization of network flow is an essential process not only for the scheduling, management, and regulation of the network's services, but also for attacks identification and for the consequent forensics' investigations. A serious potential disadvantage of the traditional software solutions used today for computer network monitoring, and specifically for the instances of effective categorization of the encrypted or obfuscated network flow, which enforces the rebuilding of messages packets in sophisticated underlying protocols, is the requirements of computational resources. In addition, an additional significant inability of these software packages is they create high false positive rates because they are deprived of accurate predicting mechanisms. For all the reasons above, in most cases, the traditional software fails completely to recognize unidentified vulnerabilities and zero-day exploitations. This paper proposes a novel intelligence driven Network Flow Forensics Framework (NF3) which uses low utilization of computing power and resources, for the Next Generation Cognitive Computing SOC (NGC2SOC) that rely solely on advanced fully automated intelligence methods. It is an effective and accurate Ensemble Machine Learning forensics tool to Network Traffic Analysis, Demystification of Malware Traffic and Encrypted Traffic Identification.

**Keywords:** network flow forensics; Security Operations Center; network traffic analysis; traffic identification; demystification of malware traffic; ensemble machine learning

---

## 1. Introduction

Network traffic analysis [1] is the method of capture, studying and analyzing network traffic flow for the purpose of performance, security and network services management. The basic strategy to network traffic analysis is a payload-based classification tactic [2] where the list of packages is sorted based on the payload, such as Mac (Layer 2), IP address (Layer 3), source/destination ports (Layer 4)

and protocols. An alternative is the statistical analysis method of the traffic behavior that is ordered based on characteristics such as interpacket arrival, session, timestamp and so on.

On the other hand, malware is a kind of malicious software used to gain access to network infrastructures without permission, to collect personal information or disrupt computer operation and facilities. It can use any event-handling procedures such as source code, dynamic scripts, or any other active content. Innovative malware types can obfuscate and remain concealed through infection and operation with sophisticated techniques that use ambiguous filenames, alteration of file features, or operation under the pretense of valid software and services to prevent investigation and deletion. Furthermore, the malicious process often tries to destabilize the entire system by bypassing the antivirus software and obfuscate active procedures, network services, and threads from suspicious URLs or registry values [3].

Moreover, the fast-growing use of encrypted traffic is changing the threat landscape. Nowadays, many services and software packages are using a type of encryption as the primary method to secure the sensitives information [4]. By the same logic, cybercriminals use advanced and highly sophisticated types of malicious operations based on progressive encryption to hide malware payload, command and control activities, or information exfiltration. Most malware types are developed to access silently and continue to exist for an extended period, to take ownership and get full control of the equipment, and to interconnect (via encryption) with the botmaster and its Command and Control (C&C) servers [4].

This paper suggests a novel network forensics framework for security operating centers that relies solely on fully adaptive computational intelligence approaches. It is an effective and accurate ensemble machine learning forensics tool that uses low utilization of computing power and resources to analyze instantly or in real time the network flow to identify encrypted or malware traffic. It is a novel cognitive analytics framework that employs an ensemble architecture which combines Support Vector Machine (SVM) [5], Artificial Neural Network (ANN) [6], Random Forest (RF) [7] and k-Nearest Neighbors (k-NN) [8] to investigate malicious activities from data flows in real time. The reason for using the ensemble technique is that complex multifactorial problems such as the one under consideration contain strong multi-variability that can be analyzed and finally solved by the sensitivity of the overlapping models. In addition, an ensemble model is appropriate to effectively express the mathematical modeling of data vectors that are used to describe complicated relationships, such as the one between normal and malicious network traffic. The combination of four different algorithms facilitates the sorting process, making each classifier more robust, and it accelerates the convergence of the generic multiple model, which is less noisy than any single one [9]. Thus, this approach offers generalization and avoids overfitting which is one of the basic targets in machine learning.

The rest of the paper is organized as follows. Section 2 presents the related work about the traffic analysis systems that have used machine learning methods. Section 3 describes the proposed NF3 model. Section 4 defines the methodology. Section 5 describes the ensemble of algorithms used by NF3. Section 6 describes the datasets. Section 7 presents the results. Section 8 contains the conclusions.

## 2. Related Work

The basic drawback of software to analyze network flows is that these applications do not offer the deep packet level details required for comprehensive analysis, as they do not have the access to each packet in the traffic flow to achieve high level application analysis. In addition, the precision of the analysis depends a portion on the sample rate selected. The higher is the sample rate, the more precise is the analysis. The type of sampling also plays a vital issue in the accuracy of outcomes. The supported sample rates are dependent on the software vendors [10].

Furthermore, all network infrastructures need to support the appropriate protocols for a comprehensive network flow analysis. Moreover, when working with many network flows, the bandwidth overhead as well as computer resources requirements for analysis procedures will have a substantial impact on the system [11].

Besides, operators are aided by visual means when analyzing big data. Their interpretation of the reality on the screen may vary due to their skills and knowledge. An inherent and implicit demand for proof of the effectiveness is to maximize operators' cyber situation awareness by adopting meaningful visualization tools as part of a comprehensive decision-support mechanism [12].

A major issue of these applications, including consistent advanced applications that rely on Deep Packet Inspection (DPI) [13] methods, is the use of signature to achieve threat identification. For this signature-based malware identification, an imperfect signature capability can be used to recognize the well-known events, provided the correct packet is sampled and the signature exists. Unfortunately, up-to-date malicious code appear that are not predictable, and these newly released forms can only be distinguished from benign files and activity by behavioral analysis or another progressive technique [14].

For example, the most recent types of malware are looking at establishing secret communications with the remote C&C servers on a systematic basis, so that cybercriminals can transfer the malicious payload to the compromised devices (bots) using hardcoded pool lists of IP addresses. Specifically, to remain hidden by the IDS/IPS, the botnets communicate using secret dynamic DNS services that are implemented in high port numbers to generate the next rendezvous-point with the botmasters. These rendezvous-points are characterized by a mixture of hundreds of random IP addresses and a very small Time-To-Live (TTL) for each partial DNS Resource Record. In addition, the use of sophisticated cryptography in malware code with the combination of the Blind Proxy Redirection (BPR) method that repeatedly redirects the requests to another group of backend servers to spoil traces and disappear the underlying networking details makes it very difficult to identify the C&C servers by law enforcement [15,16]. Hence, the botnets have become more complicated [3].

The most effective method for cyber-attacks prevention and effective investigation of malware communications is the demystification of malware traffic. Moreover, this is the primary technique to estimate the behavior of the malicious process, the intention of attacks and the degree of impairment caused by these activities [3].

The latest sophisticated malware uses the chaotic construction of the Tor network [17] to encrypt the botnet traces and modify the paths of an attack [18]. This encrypted peer-to-peer network, based on manifold layers of sophisticated encryption, complex virtual circuits and overlays that change frequently [19], certifies the secrecy among the compromised machines and the hidden services on a botnet.

Moreover, a characteristic that adds complexity in the investigation process of Tor-based malware is the fact that these types of malware operate in the transport layer of the OSI model, thus the network flow shows clients of the Secure Socket Interface (SOCKS) which operates in the session layer [20]. As a result, Tor uses port 443, so the generated traffic simulates the legitimate HTTPS traffic.

One of the most reliable methods to successfully identify the Tor-generated traffic flow is statistical analysis and the investigation of the changes in the Secure Sockets Layer (SSL) protocol [20]. For example, a statistical analysis about the related domain name, the time-to-live, etc. can identify the Tor sessions in a network full of HTTPS traffic [3,16,19,20].

NF3 is an artificial intelligence (AI) computer security technique [21–26]. Machine learning (ML) methods, using static [27] and dynamic [28] investigation to classify malicious contend [29], to achieve network traffic arrangement [30], to analyze malware traffic [31] and to identify botnets [32], has been done in the past. In contrast, numerous writers suggest different classifications methods or discovery procedures, presenting alternative classes of botnet detection [33,34]. Generally, the traffic analysis with machine learning-based methods has proved effective in the investigation of some of the biggest and most harmful cyber-attacks over the past decade [35–37].

On the other hand, Hsu et al. [38] proposed a real-time system for detecting botnets based on an anomaly detection system that inspects the delays in HTTP/HTTPS client requests with interesting outcomes. In addition, the authors of the study [39] employed several machine learning models to categorize the SSH flow, with limited features of the payload. Alshammari et al. [40] proposed an

accurate ensemble system that classifying the SSH traffic without extracting features from the payload. Holz et al. [41] investigated a precise method to trace botnets. Almubayed et al. [42] presented a method that measures the performance of several algorithms to identify the encrypted traffic in a network. Chaabane et al. [43] described an in-depth study about HTTP, BitTorrent and Tor traffic and a method to identify these protocols from user's behavior. There are several similar studies that propose methods to locate the encrypted relay nodes of the Tor network [44,45]. Mees et al. [46] developed a multi-aspect tri-dimensional picture, specific to the cyber domain, to provide a starting point calculation of the cyber situation by computing mission-specific features and related metrics with expert knowledge using fuzzy logic. Llopis et al. [12] presented a comparative analysis between visualization techniques, using the operational picture concept, to support incident handling. The mentioned authors anticipated that visualization could be enhanced by using AI algorithms to classify information and support decision making.

## 3. Proposed Framework

Since cyber systems' security is an extremely complex process, SOCs administrators cannot be based only in the use of isolated protection products installed on each checkpoint aiming to avoid an incident. The detection of an intrusion in the network should not be a manual and time-consuming process, which would offer an important advantage to the attackers. Following this point of view, the use of more effective methods of network supervision, with capabilities of automated control, are important to estimate the behavior of malware, the aim of cyber-attacks and the degree of impairment caused by malware activities.

Updating the SOC and its transformation into a NGC2SOC are also important. The ideal NGC2SOC includes advanced machine learning solutions for real-time analysis of both known and unknown threats; instant reports; data visualization tools; and other sophisticated solutions that minimize the risk of critical assets as well as fully automate the restoration of cyber security issues.

Unlike other techniques that have been proposed in the literature focused on single flow analysis approaches [17,18], the dynamic ensemble model of NF3 reduces the overfit without special requirements and computer resources.

The algorithmic approach of the proposed NF3 includes in Stage 1 a feature extraction process from network flows, as shown in the depiction of the proposed NF3 model (Figure 1). In Stage 1, these features are concurrently checked by each learning algorithm to produce an ensemble averaging model (Stage 2 in Figure 1). In an ensemble averaging model, for every instance of test dataset, the average predictions are calculated.

The analysis is about determining the normal or abnormal network traffic (network traffic analysis). If the analysis gives a positive result (Stage 3 in Figure 1) and the traffic is categorized as abnormal, it will be analyzed to specify the abnormality (demystification of malware traffic) that is taking place (Botnet, Crimeware, APT, Attack, or CoinMiner). On the contrary, normal traffic will be checked (Stage 4 in Figure 1) to identify encrypted traffic (encrypted traffic identification) and the protocol it uses (Tor, SSH, SSLweb, SSLP2P, SCP, and Skype) (Stage 5 in Figure 1) or to identify which application is being used for non-encrypted traffic (FTP, HTTP, DNS, and SMTP) (Stage 6 in Figure 1). Figure 1 is a representation of the algorithmic approach of the proposed NF3 model.

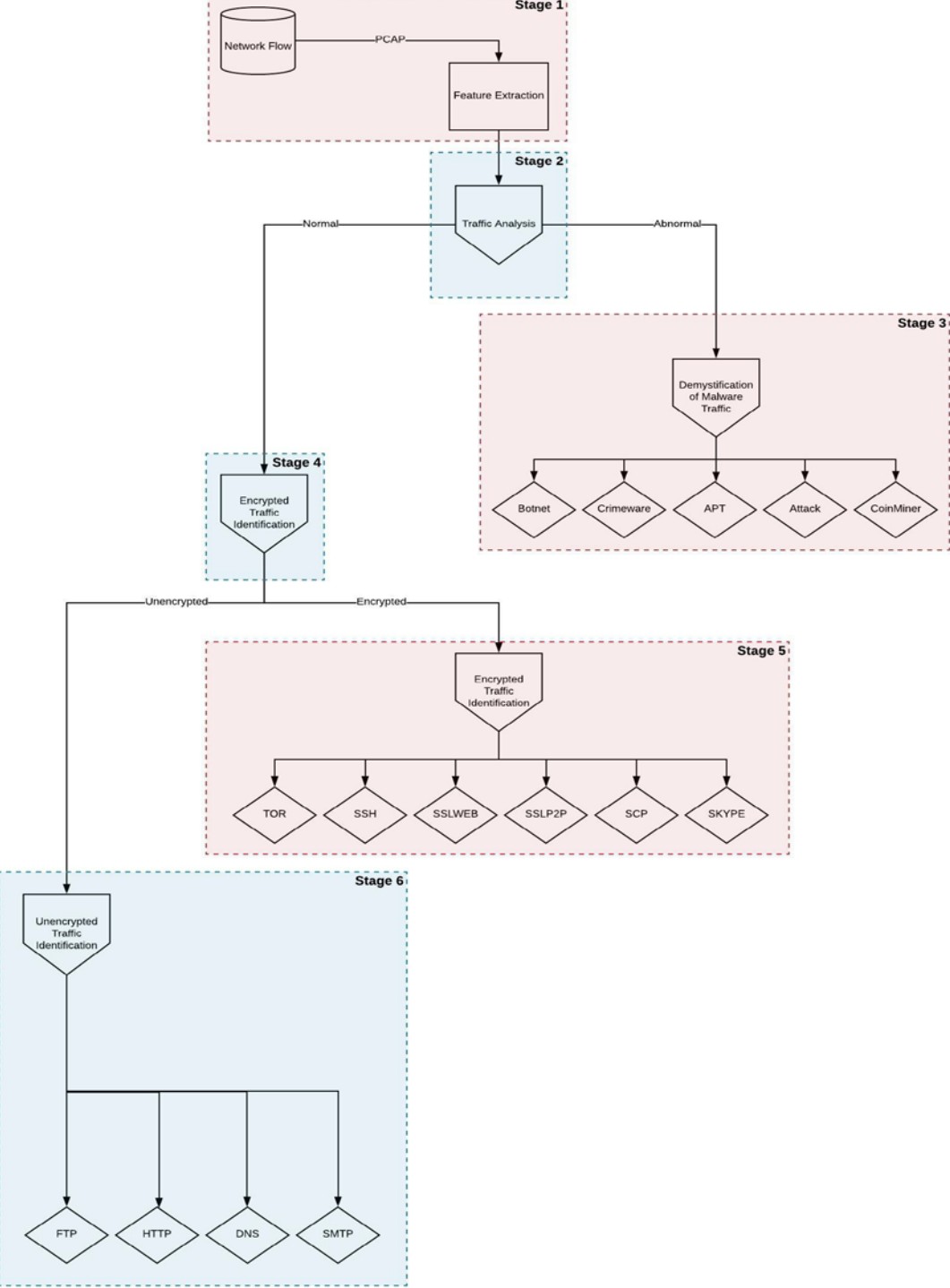

**Figure 1.** A depiction of the algorithmic approach of the proposed NF3 model.

## 4. Methodology

### 4.1. Ensemble Learning

Implementing NF3 is based on the optimal usage and the combination of reliable algorithms, which create a complete, innovative, computationally intelligent ensemble machine learning framework to solve a real cyber security problem. Ensemble approaches are meta-algorithms that syndicate numerous machine learning methods into one forecasting model to reduce variance (bagging), bias (boosting), or improve estimates (stacking) [47].

The main imperative advantages of ensemble models are that they produce more stable implementations, often improving the entire prediction, and they are capable of generalization [48]. This is a critical requirement of machine learning models so that they can adapt properly to new, previously unseen data.

An ensemble forecasting method may not necessarily yield the maximum performance, but it definitely decreases the overall risk of a particularly poorer choice. On the other hand, it should be related by a detailed examination of the elements or structure and it should explain in-depth some critical decision points correlated to the procedure and use of the approach [9,47,48]. Some of these points are presented below.

### 4.2. Ensemble Size

The number of predictors comprised in the conception of an ensemble model has a big influence on the performance of the entire model. The detailed exploration of the optimal number of classifiers is a constant research issue as well as a constant consultation among researchers, and it should be noted that there are very limited studies that address this matter [9,47,48].

In this study, comprehensive statistical research was used to determine the appropriate number of classifiers [49]. More recently, after the publishing of the "law of diminishing returns in ensemble construction" [9,47–50], it is suggested that the ideal number of classifiers that can yield the highest precision in an ensemble model for a dataset is equivalent to the obtained number of classes. However, it is generally accepted that a priori determination of the number of classifiers without scientific evidence is a precarious decision that does not guarantee the quality of the model.

To materialize the NF3 model, four classifiers were used according to "the law of diminishing returns in ensemble construction" principle. In addition, statistical tests carried out and the final ensemble size were decided with trial and error method.

### 4.3. Model Selection

The selection processes of the appropriate predictors to be comprised in an ensemble method [9,47–50] should be based on the restrictions' settings and configurations that can take into consideration different decision boundaries. For example, the obvious choice of classifiers with the smallest error in training data is considered as not proper for generating ensembles, as performance in a training dataset, even when cross validation is used, may be misleading in terms of classification performance in unknown data [51].

For this process to be effective, individual classifiers should not only display a certain level of diversity, but they should also use different operating parameters and different sets of training data, thus allowing different decision boundaries to be created, which can be combined to reduce the overall error [9,51].

In general, the selection was based on a heuristic method that considered the basic properties of how these algorithms face each situation, for instance: Are parametric models (ANN parametric, Kernel SVM non-parametric, etc.) suitable? How should outliers be handled? (For example, RF using a subgroup of training sets with bagging and subgroups of features can help reduce the effect of outliers or extreme values.) How should noise be handled? (For example, KNN has some nice properties: it is repeatedly nonlinear, it can detect linear or nonlinear dispersed data, and it tends to achieve good results with many data vectors.) The final decision was made using the statistical trial and error method.

### 4.4. Identification of the Weights of Different Models of Ensemble

An analysis that should accompany the training of an ensemble model should aim to find the optimal weights of the algorithms involved [9,47–51]. The weight vector is a very critical function in the training process of an Ensemble model, as it is used in the process of determining the reliability of predictors and the trustworthiness of their classification. In the case of higher weights, a question is

raised of how they play a more important role in defining the process of classifiers' combination and how they determine the confidence of the overall model. The usual practice of employing the same weight for all algorithms and averaging the forecasts [51,52] is a rough heuristic technique to address this challenge, which is not totally based on scientific evidence.

To create a NF3 ensemble model, the identification of the weights of different models was completely based on the statistical trial and error method described above.

### 4.5. Reliability of Ensemble

The slight difference of forecast performance in a machine learning model is one of the most important distinctive attributes for evaluating the reliability and intercity of the ensemble model. In particular, when dispersion is low and prediction solutions are consistent across multiple tests, the model is more reliable, unlike large dispersion cases, which suggest high uncertainty rates in the final forecast [52].

Ideally, the spread of the expected error should be concentrated near a value that can be described as the average error [9,47–52]. Essentially, for a prediction to be considered reliable, the observed state should behave as if it was derived from the prediction probability distribution. On the contrary, the combination of fixed categorizers for the compilation of an ensemble predictive model is a less advantageous tactic as this will not help to improve the generalization ability of the final model. Therefore, a very important reliability factor of an ensemble model is the diversity of the selected classifiers, which can be achieved with different architectures, parameter settings and training techniques.

To construct the NF3 model, various algorithms were selected based on the function method and their parameterization, which is accomplished by using different architectures, hyperparameter settings and training techniques.

### 4.6. Importance of Ensemble

Timeliness is a key issue at SOC levels, which is why they use combined cyber threat intelligence capabilities. Implementing NF3 is based on the optimal usage and combination of reliable algorithms, which create a complete ensemble machine learning framework to solve a real cyber security problem. Ensemble methods are more stable models and offer generalization. In machine learning, generalization denotes the aptitude of a model to be effective across a variety of inputs. Specifically, an ensemble model such as the proposed NF3 can fit unseen patterns such as zero-day malware or attacks. This is a major innovation that significantly improves the performance of the SOC/NOC, against sophisticated zero-day exploits.

## 5. Ensemble of Algorithms

The algorithms used and the individual determination and usage parameters of the proposed ensemble framework are briefly presented below.

### 5.1. Support Vector Machine (SVM)

SVM is mainly a classifier that creates hyperplanes in a multidimensional space that separates decision boundaries of dissimilar classes [5]. It assumes that the data are linearly separable. The SVM employs a reiterative training procedure to build an ideal hyperplane the error function is minimal when maximizing the margin subjected to a set of linear constraints. This procedure can be considered as an optimization problem that can be resolved by quadratic programming. Generally, the nonlinear data are transformed to a higher dimension to reach linear separation. For example, the kernel trick is an efficient method to transform the original data space into a high dimension that has an explicit dividing margin between classes of data. There are few tuning parameters and so the typical technique is to operate in two phases: (1) find the optimal optimization parameters; and (2) train the SVM using those parameters.

If the data are linearly separable, the decision surface has the following form [5]:

$$t_k\left(w^T x_k + b\right) \geq +1, \ k = 1, 2, 3, \ldots, N \tag{1}$$

where x is the input vector, w is the weight vector, b is bias and $w^T x + b = 0$ is the decision boundary.

When data are nonlinearly separable, which is more probable due to uncertainty, representation inaccuracy and latency, there is a classification error and the purpose of SVM is to minimize this error. A new dataset of positive numbers is inputted whose name is slack variables, and which calculate the data diversion from correct classification. In this case, the decision surface is calculated:

$$t_k\left(w^T x_k + b\right) \geq 1 - \xi_k, \ k = 1, 2, 3, \ldots, N \tag{2}$$

where $\xi_k \geq 0$ are the slack variables. Hence, we have the formulation of the SVM optimization problem which find the optimal surface $(w^*, b^*)$ with slack variables that reduce the cost of $J(w) = \frac{1}{2} w^T w$:

$$\min_{w, b} \left\{ J(w, \xi) = \frac{1}{2} w^T w + c \sum_{k=1}^{N} \xi_k \right\} \tag{3}$$

thus

$$t_k\left(w^T x_k + b\right) \geq 1 - \xi_k \text{ and } \xi_k \geq 0, \ k = 1, 2, 3, \ldots, N \tag{4}$$

where c is the capacity constant [5].

In the network traffic classification problem, the authors used the Gauss kernel SVM method to calculate the maximum-margin hyperplanes:

$$k(x, x') = \exp\left(\frac{-||x - x'||^2}{2\sigma^2}\right) \tag{5}$$

*5.2. Artificial Neural Network (ANN)*

ANNs are algorithms that simulate the human brain [6]. They are widely used for nonlinear modeling and often are characterized by computational soft computing techniques. The most common training method for ANNs algorithm is the Back-Propagation (BP) method that can be considered as a method to calculate the weights to be used in the network to minimize the error output.

The Mean Squared Error (MSE) or Root Mean Square Error (RMSE) is the performance metrics throughout the training, validation, and testing procedures of ANNs with BP training method [53,54].

$$\text{MSE} = \frac{1}{N} \sum_{i=1}^{N} (e_i)^2 = \frac{1}{N} \sum_{i=1}^{N} (t_i - a_i)^2 \tag{6}$$

$$RMSE = \sqrt{\frac{1}{n} \sum_{j=1}^{n} \left(P_{(ij)} - T_j\right)^2} \tag{7}$$

The following is a heuristic function to estimate the neurons on the hidden layer:

$$\left(\frac{2}{3} * Inputs\right) + Outputs \tag{8}$$

The Levenberg–Marquardt is the training algorithm of the MLFF ANN:

$$x_{k+1} = x_k - \left[J^T J + \mu I\right]^{-1} J^T e \tag{9}$$

where proper steps are followed to:

1.  Calculate the inputs:

$$s_j = \sum_{i=1}^{n} \left( W_{ij} X_i \right) - \theta_j(9), \ j = 1, 2, \ldots, h \tag{10}$$

2.  Calculate the output for each hidden node:

$$S_j = sigmoid_{(s_j)} = \frac{1}{(1 + exp(-s_j))}, \ j = 1, 2, \ldots, h \tag{11}$$

3.  Calculate the overall outputs:

$$o_k = \sum_{j=1}^{h} \left( W_{jk} S_j \right) - \theta'_k(11), \ k = 1, 2, \ldots, m \tag{12}$$

$$O_k = sigmoid_{(o_k)} = \frac{1}{(1 + exp(-o_k))}(12), \ k = 1, 2, \ldots, m \tag{13}$$

### 5.3. Random Forest (RF)

The Random Forests (RF) is a forecasting method that operates by creating a plethora of decision trees. The training method for RF applies a general bootstrap aggregating method, or a bagging technique, or a tree-learning process. Generally, the RF algorithm can be described as follows [7]:

(1)  Draw n tree bootstrap samples from the source dataset.
(2)  At each node of the bootstrap samples, grow an unpruned predictor tree and choose the best split between variables.
(3)  Predict new data by aggregating the predictions of the n trees and estimate the error at each iteration using the "out-of-bag" method.

### 5.4. k-Nearest Neighbors (k-NN)

The K-Nearest Neighbors (k-NN) algorithm (also known as instance-based learning) is a simple classification method developed to meet the need for discriminant analysis when dependable parametric estimations of likelihood concentrations are unidentified or difficult to regulate [8]. To forecast a new data point, the nearby *k* neighbors are determined in the training set and then they follow a voting process to produce the final prediction. *k* is a parameter defined by the user and an unlabeled data-vector is classified with the most frequent label between the *k* training samples nearest to that request point. To determine the "nearest neighbors" the Euclidean distance function between the testing and training samples is employed. The Euclidean distance is defined as [53,54]:

$$dist_{Euklidean} \left( x_0, x_j \right) = \sqrt{\sum_{i=1}^{n} \left( x_0^i - x_j^i \right)^2} \tag{14}$$

The k-NN algorithm can be summarized as follows:

1.  A positive integer *k* is definite, along with a new sample.
2.  The closest *k* entries are selected.
3.  The most usual classification of these entries is determined and given to the new sample.

The error probability of the model is calculated by the following equation:

$$P_B \leq P_{k-NN} \leq P_B + \frac{1}{\sqrt{ke}} \tag{15}$$

where $P_B$ is the optimal Bayesian error (minimum when $k \to \infty$).

## 6. Datasets

### 6.1. Features Extraction

The feature extraction process from the network flow was based on the theoretical background of the way in which the TCP protocol work and moreover on the dependable submission between the network and the application layers of the TCP header structure, the three-way handshake method and the communications security over the SSL protocol [55]. Our research team carried out an extensive investigation to find the most effective independent variables that describe with maximum correlation and precision the problem of network traffic analysis under the strict condition of low utilization of computing power and resources. This determination resulted in the construction of effective datasets, able to produce an accurate framework that can adapt properly to new data.

The features management and the extraction process of the set of 46 features, including all of the network flows, is analytically described in [56]. It should be emphasized that this feature extraction process is also enriched by some novel representation techniques for simple structures and data modification programmed in the Python programming language.

### 6.2. Data

The following five datasets have been developed to produce highly multifaceted scenarios that can be perceived in a network traffic flow and which are appropriate for the training of the proposed NF3 model.

Firstly, the Network Traffic Analysis (NTA) binary dataset contains 30 independent variables and two classes (normal or abnormal). This dataset contains 208,629 instances (119,287 normal samples chosen from the Pcaps which are packet captures obtained from an application programming interface for capturing network traffic) from National Cyber Watch Mid-Atlantic Collegiate Cyber Defense Competition and 39,342 abnormal samples chosen from the Contagio Malware Dump [57].

Secondly, the Demystification of Malware Traffic (DMT) multiclass dataset comprises 30 independent variables and five malware classes (Botnet, Crimeware, APT, Attack and CoinMiner). This dataset contains 168,501 instances chosen from [57] including Pcaps files that captured malware traffic from honeypots, sandboxes and real-world intrusions.

Thirdly, Encrypted Traffic Analysis (ETI) binary dataset comprises 30 independent variables and two classes (encrypted or non-encrypted). This dataset contains 166,874 instances (93,024 encrypted and 73,850 unencrypted) from the "Inter-Service Academy Cyber Defense Competition" served by Information Technology Operations Center (ITOC), United States Military Academy (West Point, NY, USA) [58].

Fourthly, Encrypted Traffic Identification (EnTI) multiclass dataset comprises 30 independent variables and six classes that represent encrypted protocols (Tor, SSH, SSLweb, SSLP2P, SCP, and Skype). This dataset contains 214,155 instances from [59] including a list of Pcaps file repositories, which are freely available on the Internet.

Finally, Unencrypted Traffic Identification (UTI) multiclass dataset comprises 30 independent variables and four classes of unencrypted network protocols (FTP, HTTP, DNS, and SMTP). This dataset contains 214,155 instances from [59].

The full list of the 30 data features is detailed in [60].

## 7. Results

In the case of multi-class or binary classification, the estimation of the actual error requires the probability density of all categories [53,54]. The classification accuracy is estimated by the employment of a Confusion Matrix (CM) or error matrix that is a detailed matrix that allows visualization of the performance of a model. A Receiver Operating Characteristic (ROC) curve is a

graph showing the performance of a classification model at all classification thresholds. The number of misclassifications is related to the False Positive (FP) and False Negative (FN) indices appearing in the confusion Matrix. FP is the case where a positive result is wrongly received, and FN is exactly the opposite. In addition, a True Positive (TP) is a correctly received positive result. A True Negative (TN) is correctly indicating the condition being tested is not present. The ROC curve plots two parameters: True Positive Rate (TPR), also known as Sensitivity (Equation (15)), and True Negative Rate (TNR), also known as Specificity (Equation (16)) [53,54]. The Total Accuracy (TAC) is defined using Equation (17) [53,54]:

$$\text{TPR} = \frac{\text{TP}}{\text{TP} + \text{FN}} \tag{16}$$

$$\text{TNR} = \frac{\text{TN}}{\text{TN} + \text{FP}} \tag{17}$$

$$\text{TAC} = \frac{\text{TP} + \text{TN}}{\text{N}} \tag{18}$$

The Precision (PRE), Recall (REC) and F-Score indices are defined in Equations (18)–(20) [53,54]:

$$\text{PRE} = \frac{\text{TP}}{\text{TP} + \text{FP}} \tag{19}$$

$$\text{REC} = \frac{\text{TP}}{\text{TP} + \text{FN}} \tag{20}$$

$$\text{F} - \text{Score} = 2 \times \frac{\text{PRE} \times \text{REC}}{\text{PRE} + \text{REC}} \tag{21}$$

The following tables presents an extensive comparison between algorithms.

**Table 1.** Comparison between algorithms.

| Network Traffic Analysis (Binary) (208.629 Instances) | | | | | | |
|---|---|---|---|---|---|---|
| **Classifier** | **Classification Accuracy & Performance Metrics** | | | | | |
| | **TAC** | **RMSE** | **PRE** | **REC** | **F-Score** | **ROC_Area** |
| SVM | 98.01% | 0.1309 | 0.980 | 0.980 | 0.980 | 0.980 |
| MLFF ANN | 98.13% | 0.1295 | 0.981 | 0.981 | 0.981 | 0.994 |
| k-NN | 96.86% | 0.1412 | 0.970 | 0.970 | 0.970 | 0.970 |
| RF | 97.12% | 0.1389 | 0.972 | 0.971 | 0.971 | 0.971 |
| Ensemble | 97.53% | 0.1351 | 0.976 | 0.975 | 0.975 | 0.979 |

**Table 2.** Comparison between algorithms.

| Demystification of Malware Traffic (Multiclass) (168.501 Instances) | | | | | | |
|---|---|---|---|---|---|---|
| **Classifier** | **Classification Accuracy & Performance Metrics** | | | | | |
| | **TAC** | **RMSE** | **PRE** | **REC** | **F-Score** | **ROC_Area** |
| SVM | 96.63% | 0.1509 | 0.967 | 0.967 | 0.968 | 0.970 |
| MLFF ANN | 96.50% | 0.1528 | 0.981 | 0.981 | 0.981 | 0.965 |
| k-NN | 94.95% | 0.1602 | 0.970 | 0.970 | 0.970 | 0.950 |
| RF | 95.91% | 0.1591 | 0.972 | 0.971 | 0.971 | 0.960 |
| Ensemble | 95.99% | 0.1557 | 0.972 | 0.972 | 0.973 | 0.961 |

**Table 3.** Comparison between algorithms.

| Classifier | Encrypted Traffic Analysis (Binary) (166.874 Instances) | | | | | |
|---|---|---|---|---|---|---|
| | Classification Accuracy & Performance Metrics | | | | | |
| | TAC | RMSE | PRE | REC | F-Score | ROC_Area |
| SVM | 98.99% | 0.1109 | 0.989 | 0.990 | 0.990 | 0.990 |
| MLFF ANN | 99.12% | 0.1086 | 0.998 | 0.998 | 0.998 | 0.998 |
| k-NN | 97.84% | 0.1372 | 0.975 | 0.975 | 0.978 | 0.980 |
| RF | 98.96% | 0.1107 | 0.989 | 0.989 | 0.989 | 0.990 |
| Ensemble | 98.72% | 0.1168 | 0.987 | 0.987 | 0.988 | 0.989 |

**Table 4.** Comparison between algorithms.

| Classifier | Encrypted Traffic Identification (Multiclass) (214.155 Instances) | | | | | |
|---|---|---|---|---|---|---|
| | Classification Accuracy & Performance Metrics | | | | | |
| | TAC | RMSE | PRE | REC | F-Score | ROC_Area |
| SVM | 90.31% | 0.1906 | 0.905 | 0.905 | 0.906 | 0.950 |
| MLFF ANN | 92.67% | 0.1811 | 0.930 | 0.930 | 0.928 | 0.960 |
| k-NN | 85.19% | 0.2032 | 0.890 | 0.890 | 0.890 | 0.935 |
| RF | 91.56% | 0.1800 | 0.920 | 0.916 | 0.916 | 0.930 |
| Ensemble | 89.93% | 0.1887 | 0.911 | 0.910 | 0.910 | 0.943 |

**Table 5.** Comparison between algorithms.

| Classifier | Unencrypted Traffic Identification (Multiclass) (186.541 Instances) | | | | | |
|---|---|---|---|---|---|---|
| | Classification Accuracy & Performance Metrics | | | | | |
| | TAC | RMSE | PRE | REC | F-Score | ROC_Area |
| SVM | 99.92% | 0.1003 | 0.999 | 0.999 | 0.999 | 0.999 |
| MLFF ANN | 99.91% | 0.1008 | 0.999 | 0.999 | 0.999 | 0.999 |
| k-NN | 98.98% | 0.1020 | 0.989 | 0.989 | 0.990 | 0.995 |
| RF | 99.93% | 0.1001 | 0.999 | 0.999 | 0.999 | 0.999 |
| Ensemble | 99.68% | 0.1008 | 0.996 | 0.996 | 0.997 | 0.998 |

## 8. Conclusions

### 8.1. Discussion

As shown in the above tables, the ensemble method appears to have the same or a slightly lesser performance across all datasets, compared to the winner (more accurate) algorithm. This fact does not detract in any case from the value of the proposed method considering that the proposed ensemble processing approach builds a robust predictive model that reduces the overfit. As stated by this reasonable analysis, it seems that this method is an appropriate method for complex multifactorial problems such as the one under consideration.

High precision shows the rate of positive predictions is precise, whereas high recall specifies the rate of positive events is correctly predicted. Precision is also a measure of correctness or quality, whereas recall is a degree of completeness or quantity. In all cases, the proposed model had high average precision and very high recall, meaning the ensemble method is a robust and stable method that returns substantial results. Correspondingly, the F-Score works greatest if false positives and false negatives have a comparable cost. In other words, the F-Score is the harmonic average of the precision and recall, and, in all scenarios, the proposed ensemble method score reaches its best value near 1 (perfect precision and recall). Finally, the high ROC area values of the method provide details on class distribution and it is related to a cost or benefit analysis of an indicative decision making.

Tables 1–5 clearly show that the ensemble model is a quite promising method considering that it offers comparable prediction and supplementary stable models, as the overall behavior of a multiple model is less noisy than a corresponding single one. It is important to say that analyzing and identifying some parameters that can determine a type of threat such as cyber-attacks is a partly subjective, nonlinear and dynamic process. The proposed NF3 for the NGC2SOC may use novel representation techniques such as 3D-models or immersive visualization through the use of Virtual Reality (VR) glasses [12] since these techniques allow operators a profound analysis when confronted with information-overwhelmed situations. Additionally, these techniques incorporate interfaces with available data sources, e.g., Malware Information Sharing Platform (MISP).

*8.2. Innovation*

The most significant innovation of NF3 is the proposed creation of a next generation cognitive computing SOC which will use machine learning technologies to adapt to different contexts with minimal human supervision. This new approach has the potential to help organizations combat cybercriminals in real-time. Moreover, traying an all-inclusive analysis of the NF3 model, authors obviously comprehend that, in the proposed approach, the identification of malicious software or attacks is done in unproductive time, before troubling or disrupting the operation of the entire infrastructure. This is a major innovation that generates new standpoints in the implementation of the IDS/IPS, which adopt intelligent protection against innovative zero-day vulnerabilities. The proposed framework adds a higher integrity to the security infrastructures improving its cyber resilience with high identification speed, ease of implementation, minimal human intervention, and minimal computational resources for network traffic analysis, demystification of malware traffic and encrypted traffic identification. In addition, the feature extraction and selection procedure is very interesting and innovative. This feature has occurred after comprehensive research about the network protocols work in the lower and upper layers of the OSI model. It is important also to highlight that the datasets occurred after evaluations concerning the boundaries and the determination of normal or abnormal behavior of the network procedures.

Finally, an impressive innovation of the proposed framework is the ability to identify DoS/DDoS attacks with high precision. A DoS/DDoS [61–63] attack is orchestrated by creation of high rate malicious traffic using sources and services of compromised machines establishing multiple simultaneous connections. One of the most important characteristics of this traffic is the modification in the number of packets flows in a time-window. For example, the statistical analysis of the packet count can be used to detect DoS/DDoS attacks. As detailed in [60], the dataset includes features such as the total number of packets traveling in the flow in a time-window; the minimum, maximum and average packet length; the minimum, maximum and average interarrival time between two packets; and the time elapsed from the first packet to the last packet. All these features are useful to identify anomalies in the network flow related to DoS/DDoS attacks.

*8.3. Synopsis*

This research paper proposes NF3, an innovative, reliable and highly effective network forensics tool, employing computational intelligence principles. It is a highly suitable method in cases where the traditional signature-based applications are computationally infeasible. It is an ensemble machine learning framework that is based on the optimal combination of four highly efficient and fast learning algorithms that create a comprehensive intelligent cyber security system proposed for the next generation cognitive computing SOC. This sophisticated application, combined with the promising results that have emerged, constitutes a credible innovative proposal for the standardization and design of improved cyber security infrastructures. Moreover, this implementation is done by using datasets that respond to specialized, realistic scenarios. In addition, this framework implements a data analytics approach that attempts to balance latency, throughput, and fault tolerance using integrated and accurate views of new entrant data flows. It is important to mention that this paper

proposes a novel intelligence driven network flow forensics framework which uses low computing utilization resources to network traffic analysis, demystification of malware traffic and encrypted traffic identification. Given the data dimensionality, it seems suitable for most existing deep learning solutions, but deep learning methods are extremely computationally expensive in the training process and are very time-consuming. For example, on a deep convolutional network, the training procedure can take several days. The most sophisticated models need to spend much time to train using equipped with expensive GPUs. In contrast, the proposed NF3 model can take few minutes to train completely from scratch. In addition, the determination of the hyperparameters, topology, training methods, etc. is a black box and is difficult to comprehend. The classifiers used in the NF3 framework make it much easier to handle data as well as to understand the architecture which uses few system resources to train or retrain the models.

*8.4. Future Works*

Future research could involve a further analysis of the ensemble framework under a hybrid structure, which will handle many data using batch and stream processing methods (lambda architecture). In addition, semi-supervised methods algorithms and online learning algorithms methods could be used to extract and manipulees hidden knowledge between the inhomogeneous data that arise in network flow analysis. In addition, NF3 could be enhanced by further optimizing the parameters of the ensemble framework, so that an even more effective, precise, and faster classification process could be reached. Customized visualization incorporated into the proposed NF3 would assist SOC operators in understanding the cyber situation. Multi-format representations may support a reporting system as part of an overall decision mechanism. In addition, it would be important to study the expansion of this system by implementing the same architecture in a parallel and distributed big data analysis system such as Hadoop. Finally, a supplementary element that could be considered in the way of future expansion concerns the operation of NF3 with self-adaptive improvement and meta-agnostic-learning methods to fully automate the defense against sophisticated cyber-attacks.

**Author Contributions:** Conceptualization, K.D. and P.K.; Investigation, K.D.; Methodology, K.D. and P.K.; Software, K.D. and L.I.; Validation, K.D., P.K., N.T., S.L.S. and L.I.; Formal Analysis, K.D., P.K., N.T., S.L.S. and L.I.; Resources, K.D. and L.I.; Data Curation, K.D., P.K., N.T. and L.I.; Writing—Original Draft Preparation, K.D.; Writing—Review and Editing, K.D., P.K., N.T., S.L.S. and L.I.; and Supervision, P.K.

**Funding:** This research received no external funding.

**Conflicts of Interest:** The authors declare no conflict of interest.

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
