# Peer review of "The Next Generation Cognitive Security Operations Center: Network Flow Forensics Using Cybersecurity Intelligence"

_2504-2289, doi:10.3390/bdcc2040035_

Reviewer 1 Report

The manuscript proposes a novel intelligence driven Network Flow Forensics Framework (NF3) which uses minimum computational power and resources, for the Next Generation Cognitive Computing SOC (NGC2SOC) that rely solely on advanced fully automated intelligence methods. 

The novelty of the approach is questionable, as it only uses existing ML techniques and applies them. The proposed NF3 tool in interesting from a professional aspect, but questionable as a novel contribution. 

The novelty of the approach needs to be corroborated. Additionally, a conclusion to sum up the results could be given, not only a discussion, followed by a future work section. Therefore the presentation part of the manuscript need improvement. 

Author Response

Dear Reviewer

We deeply appreciate the time and effort you have spent in reviewing our manuscript. Your comments are very helpful for revising and improving our paper much further. We are providing the answers to your comments below.

Cordially

Konstantinos Demertzis, Panayiotis Kikiras, Nikos Tziritas, Salvador Llopis Sanchez, Lazaros Iliadis

1)      The novelty of the approach is questionable, as it only uses existing ML techniques and applies them. The proposed NF3 tool in interesting from a professional aspect, but questionable as a novel contribution. The novelty of the approach needs to be corroborated.

Ans.1 Thank you for careful reading. The paper is an applied research paper that deals with solving practical problems such as a critical cybersecurity issue. According to the scope of the Journal that is something highly significant and of particular value. In addition, scientifically, the originality and contribution of the paper are described in the new section 8.2 “Innovation”.

2)      Additionally, a conclusion to sum up the results could be given, not only a discussion, followed by a future work section.

Ans.2 Thank you for this helpful comment. We have added section 8 Conclusions where includes a discussion of the results, the innovations of the proposed framework, a synopsis and future works.

3)      Therefore, the presentation part of the manuscript needs improvement.

Ans.3 We would like to thank the reviewer for this constructive comment. We have rearranged the entire paper and have improved a lot the presentation and the background of the entire manuscript. Now the paper is self-consistent.

Reviewer 2 Report

The authors propose a Network Flow Forensics Framework which uses minimum computational power and resources, for the Next Generation Cognitive Computing SOC that rely on advanced fully automated intelligence methods. The topic of the paper is not well-aligned with the scope of the journal and the paper is poorly written and very difficult to follow. The authors have not stated clearly their contributions and they provide a very confusing literature review, citing 58 (!) research works without practically explaining in detail what the other researchers have already achieved and proposed in the recent literature regarding the examined research topic. The authors mix together so many notions from the recent literature which are hot potatoes, such as Machine Learning, Support Vector Machine, Cognitive Computing, Artificial Neural Networks, etc.,  giving the impression of a listing of buzzwords which are not connected with each other. Furthermore, in figure 1, the authors illustrate the algorithmic approach that is adopted for the proposed Network Flow Forensics Framework, which is totally not understandable and not supported by the provided description. This makes the reader unable to follow the analysis of the authors from the beginning of the paper. Moreover, the manuscript lacks of any theoretical contribution and simply exploits very well-known concepts from the literature, such as the Support Vector Machine, Artificial Neural Networks, random forest, k-nearest neighbors, etc. In Section 4, there is not new or extension-oriented contributions to the already existing approaches in the literature. Also, the usage of English language is very poor and the authors should check the whole manuscript before resubmitting it to another venue.

Author Response

Dear Reviewer

We deeply appreciate the time and effort you have spent in reviewing our manuscript. Your comments are very helpful for revising and improving our paper much further. We are providing the answers to your comments below.

Cordially

Konstantinos Demertzis, Panayiotis Kikiras, Nikos Tziritas, Salvador Llopis Sanchez, Lazaros Iliadis

1)      The topic of the paper is not well-aligned with the scope of the journal and the paper is poorly written and very difficult to follow.

Ans1. Thank you for careful reading. The paper is an applied research paper that deals with a practical real cybersecurity problem. Applied research is the practical application of science. It accesses and uses accumulated theories, knowledge, methods, and techniques, for a specific purpose. According to the scope and the aims of the Journal that is something highly significant and of particular value. Also, we have rearranged the entire paper and have improved a lot the presentation and the background of the entire manuscript. Now the paper is self-consistent.

2)      The authors have not stated clearly their contributions and they provide a very confusing literature review, citing 58 (!) research works without practically explaining in detail what the other researchers have already achieved and proposed in the recent literature regarding the examined research topic.

Ans.2 Thank you for this constructive comment. We have rearenged the Related Work section of the paper and we have discussed the literature review thoroughly.

3)      The authors mix together so many notions from the recent literature which are hot potatoes, such as Machine Learning, Support Vector Machine, Cognitive Computing, Artificial Neural Networks, etc., giving the impression of a listing of buzzwords which are not connected with each other.

Ans.3 We would like to thank the reviewer for this comment that gives us the chance to clarify things further. In Machine Learning, ensemble methods use multiple learning algorithms to obtain better predictive performance than could be obtained from any of the constituent learning algorithms alone. The classifiers that used (SVM, ANN, k-NN and RF) in the NF3 framework make it a robust and accurate model. In general, the term Cognitive Computing has been used to refer to new hardware and/or software that mimics the functioning of the human brain and helps to improve human decision-making. This applied research is a reliable and highly effective network forensics tool, employing computational intelligence principles, that improve human decision-making in the cybersecurity.

4)      Furthermore, in figure 1, the authors illustrate the algorithmic approach that is adopted for the proposed Network Flow Forensics Framework, which is totally not understandable and not supported by the provided description. This makes the reader unable to follow the analysis of the authors from the beginning of the paper.

Ans.4 We have changed the figure and we have discussed this figure in section 3 thoroughly. The depiction of the proposed model reads much better now. Thank you for this helpful comment.

5)      Moreover, the manuscript lacks any theoretical contribution and simply exploits very well-known concepts from the literature, such as the Support Vector Machine, Artificial Neural Networks, random forest, k-nearest neighbors, etc. In Section 4, there is not new or extension-oriented contributions to the already existing approaches in the literature.

Ans.5 Thank you for this valuable comment. The originality and contribution of the paper are described in the new section 8.2 “Innovation”.

6)      Also, the usage of English language is very poor, and the authors should check the whole manuscript before resubmitting it to another venue.

Ans.6 Thank you for the remarks and for the careful reading. We have rearranged the entire paper and have improved the usage of the English language of the entire manuscript. The paper reads much better now, and the work presented has improved to a level acceptable for the readership and the scientific standing of this journal.

Reviewer 3 Report

The paper introduces a novel intelligence driven Network Flow Forensics Framework (NF3) which uses minimum computational power and resources, that rely solely on advanced fully automated intelligence methods. It is achieved by taking advantage of ensemble machine learning. In general terms the contribution is interesting and well-presented, but this reviewer suggests the following modifications prior to its publications.

The introduction format is confusing. It should present a unique section (i.e. Introduction) that provides the motivation, summarizes the main contributions of the publication, and explains the organization of the rest of the document. Current SoA subsections within Introduction should be moved to Section 2 ”Related Works”.

Algorithms described at the proposal section should be move to a particular Section (SVM, RF, ANN, etc.), in this way separating the paper contributions from previous work.

Given the data dimensionality, it seems suitable for most of the existing Deep Learning solutions. Why this contribution may outperfor them?

This reviewer suggests to include previous publications in the comparative tables. There are already related works evaluated at the same datasets.

The timeliness is a key issue at SOC/NOC levels, that’s why they use to combine cyber threat intelligent capabilities. What is the expected performance of the proposal

Author Response

Dear Reviewer

We deeply appreciate the time and effort you have spent in reviewing our manuscript. Your comments are very helpful for revising and improving our paper much further. We are providing the answers to your comments below.

Cordially

Konstantinos Demertzis, Panayiotis Kikiras, Nikos Tziritas, Salvador Llopis Sanchez, Lazaros Iliadis

1)      The introduction format is confusing. It should present a unique section (i.e. Introduction) that provides the motivation, summarizes the main contributions of the publication, and explains the organization of the rest of the document. Current SoA subsections within Introduction should be moved to Section 2 ”Related Works”.

Ans.1 Thank you for this constructive comment. We have reorganized the Introduction section of the paper.

2)      Algorithms described at the proposal section should be move to a particular Section (SVM, RF, ANN, etc.), in this way separating the paper contributions from previous work.

Ans.2 We would like to thank the reviewer for this comment. We have added section 5 “Ensemble of Algorithms” that described the proposed algorithms of the Ensemble framework.

3)      Given the data dimensionality, it seems suitable for most of the existing Deep Learning solutions. Why this contribution may outperform them?

Ans.3 We would like to thank the reviewer for this comment that gives us the chance to clarify things further. This paper proposes a novel intelligence driven network flow forensics framework which uses minimum computational power and resources to network traffic analysis, demystification of malware traffic and encrypted traffic identification. The deep learning methods are extremely computationally expensive in the training process. The most sophisticated models need to spend shedloads of time to train using equipped with expensive GPUs. Also, the determination of the hyperparameters, topology, training methods, etc.  is a black box and is difficult to comprehend. The classifiers that used in the NF3 framework make it much easier to handle data, to understanding the architecture that used and uses minimum computational power and resources to train or retrain the models. We have mentioned this matter in the Synopsis section.

4)      This reviewer suggests including previous publications in the comparative tables.

There are already related works evaluated at the same datasets.

Ans.4 We would like to thank the reviewer for this comment that gives us the chance to clarify things further. The datasets used in this study was created exclusively for this research paper and has emerged after extensive research in the way in which the network protocols work in the OSI layers of the system. It is important also to notice that the dataset occurred after comparisons regarding the boundaries and the determination of normal or abnormal behaviour of the network protocols. We have discussed this matter in section Innovation thoroughly.

5)      The timeliness is a key issue at SOC/NOC levels, that’s why they use to combine cyber threat intelligent capabilities. What is the expected performance of the proposal

Ans.5 This is a very interesting comment. Implementing NF3 is based on the optimal usage and the combination of reliable algorithms, which create a complete ensemble machine learning framework in order to solve a real cyber security problem. Ensemble methods are meta-algorithms that combine several machine learning techniques into one predictive model in order to decrease variance, bias or improve predictions. These are more stable models, as the overall behavior of a multiple model is less noisy than a corresponding single one and offer generalization. In machine learning, generalization denotes to the aptitude of a model to be effective across a variety of inputs. Specifically, an ensemble model such as the proposed NF3 has the ability to fit unseen patterns like zero-days malware or attacks. This is a major innovation that significantly improves the performance of the SOC/NOC, against sophisticated zero-day exploits. We have discussed this important matter thoroughly in the new 4.6 section “Importance of Ensemble”.

Round  2

Reviewer 1 Report

The paper was much improved compared to the original version and most issues raised by the reviewers were addressed. However, there are still some parts that need to be improved:

Some comparison to related methods which address the same issue as the presented framework have similar accuracy". It should be further highlighted why he presented NF3 model is superior to competitive approaches.

In the light of the previous comments also a comparison in regard to "performance" could be given - what is the computational load to train the NF3 model and how does it compare to competitive methods and/or solutions.

Author Response

Dear Reviewer

We deeply appreciate the time and effort you have spent in reviewing our manuscript. Your comments are very helpful for revising and improving our paper much further. We are providing the answers to your comments below.

Cordially

Konstantinos Demertzis, Panayiotis Kikiras, Nikos Tziritas, Salvador Llopis Sanchez, Lazaros Iliadis

1)      Some comparison to related methods which address the same issue as the presented framework have similar accuracy". It should be further highlighted why he presented NF3 model is superior to competitive approaches.

Ans.1 T We would like to thank the reviewer for this constructive comment that gives us the chance to clarify things further. The datasets used in this study was created exclusively for this research paper and has emerged after extensive research in the way in which the network protocols work in the OSI layers of the system. We have discussed this matter in section “Innovation” thoroughly.

The proposed ensemble method appears to have the same or a slightly lesser performance across all datasets, compared to the winner (more accurate) algorithm. This fact does not detract in any case from the value of the proposed method taking into account that the proposed ensemble processing approach builds a predictive model that reduces the overfit. Generally, the superiority of the NF3 framework focuses on the robustness, accuracy, and generalization ability that offer, as the overall behavior of the ensemble model is less noisy than a corresponding single one (for example the more accurate algorithm). Specifically, the proposed ensemble NF3 model reduce overfitting, decrease variance or bias, and without reduce significant the precision of the model can fit unseen patterns like zero-day vulnerabilities or other sophisticated attacks. This is a major innovation that significantly improves the overall reliability of the model. We have discussed this important matter thoroughly in the sections “Importance of Ensemble”, “Innovation” and “Synopsis”.

2)      In the light of the previous comments also a comparison in regard to "performance" could be given - what is the computational load to train the NF3 model and how does it compare to competitive methods and/or solutions.

Ans.2 It is important to mention that this paper proposes a novel network flow forensics framework which uses low utilization of computing power and resources. Given the data dimensionality, it seems suitable for most of the existing Deep Learning solutions, but the deep learning methods are extremely computationally expensive in the training process and are very time-consuming. For example, on a deep convolutional network, the training procedure can take several weeks. The most sophisticated models need to spend shedloads of time to train using equipped with expensive GPUs. In contrast, the proposed NF3 model can take a few minutes to train completely from scratch. We have added in the “Synopsis” section an explanation and comparison about the performance of the proposed model. Thank you for this helpful comment.

Reviewer 2 Report

The authors propose a novel intelligence driven Network Flow Forensics Framework (NF3) which uses minimum computational power and resources, for the Next Generation Cognitive Computing SOC (NGC2SOC) that rely solely on advanced fully automated intelligence methods. The paper is overall well-written and it seems that it is a revision from a previous review round as the authors have highlighted several sections, which practically shows that they have almost rewritten the whole manuscript. The topic of the paper is very-interesting. The authors have clearly stated their contributions and the provided analysis is concrete and correct. Also, the provided numerical results are rich and detailed, showing the contributions, applicability and novelty of the proposed framework. My main question regarding this manuscript that the authors seem to neglect is how the proposed framework can address the denial of service attacks, the distributed denial of service attacks (e.g., Sagduyu, Y.E. and Ephremides, A., 2009. A game-theoretic analysis of denial of service attacks in wireless random access. Wireless Networks, 15(5), pp.651-666, Sagduyu, Yalin Evren, Randall A. Berryt, and Anthony Ephremidesi. "Wireless jamming attacks under dynamic traffic uncertainty." In Modeling and Optimization in Mobile, Ad Hoc and Wireless Networks (WiOpt), 2010 Proceedings of the 8th International Symposium on, pp. 303-312. IEEE, 2010.)and the interference jamming (e.g., Tsiropoulou, E.E., Baras, J.S., Papavassiliou, S. and Qu, G., 2016, November. On the Mitigation of Interference Imposed by Intruders in Passive RFID Networks. In International Conference on Decision and Game Theory for Security (pp. 62-80). Springer, Cham), which all of them are part of the cybersecurity threats. The authors are encouraged to clarify this point and provide additional discussion in the related work. Also, a minor comment is that the authors should carefully check the whole manuscript regarding the usage of the English language, as it needs improvement.

Author Response

Dear Reviewer

We deeply appreciate the time and effort you have spent in reviewing our manuscript. Your comments are very helpful for revising and improving our paper much further. We are providing the answers to your comments below.

Cordially

Konstantinos Demertzis, Panayiotis Kikiras, Nikos Tziritas, Salvador Llopis Sanchez, Lazaros Iliadis

1)      My main question regarding this manuscript that the authors seem to neglect is how the proposed framework can address the denial of service attacks, the distributed denial of service attacks (e.g., Sagduyu, Y.E. and Ephremides, A., 2009. A game-theoretic analysis of denial of service attacks in wireless random access. Wireless Networks, 15(5), pp.651-666, Sagduyu, Yalin Evren, Randall A. Berryt, and Anthony Ephremidesi. "Wireless jamming attacks under dynamic traffic uncertainty." In Modeling and Optimization in Mobile, Ad Hoc and Wireless Networks (WiOpt), 2010 Proceedings of the 8th International Symposium on, pp. 303-312. IEEE, 2010.) and the interference jamming (e.g., Tsiropoulou, E.E., Baras, J.S., Papavassiliou, S. and Qu, G., 2016, November. On the Mitigation of Interference Imposed by Intruders in Passive RFID Networks. In International Conference on Decision and Game Theory for Security (pp. 62-80). Springer, Cham), which all of them are part of the cybersecurity threats. The authors are encouraged to clarify this point and provide additional discussion in the related work.

Ans.1 We would like to thank the reviewer for this comment. A DoS or DDoS attack is orchestrated by creation high rate malicious traffic using sources and services of compromised machines establishing multiple simultaneous connections. One of the most important characteristics of this traffic is the modification in the number of packets flows in a time-window. For example, the statistical analysis of the packet count is one of the most important properties that it can be used to detect DoS/DDoS attacks. The features of the datasets used in this research, are proper to identify anomalies in the network flow that related to DoS/DDoS attacks. Specifically, the datasets include features such as the total number of packets traveling in the flow in a time-window, the minimum, maximum and average packet length, the minimum, maximum and average interarrival time between two packets and the time elapsed from the first packet to the last packet. We have added in the “Innovation” sections an additional explanation of how the proposed framework can address the DoS and DDoS attacks.

 2)      Also, a minor comment is that the authors should carefully check the whole manuscript regarding the usage of the English language, as it needs improvement.

Ans.2 Thank you for the remarks and for the careful reading. We have rearranged the entire paper and have improved the usage of the English language of the entire manuscript. The paper reads much better now, and the work presented has improved to a level acceptable for the readership and the scientific standing of this journal.

Round  3

Reviewer 1 Report

The paper was much improved compared to the original version and all issues raised by the reviewers were addressed.